# Area per player in small-sided games to replicate the external load and estimated physiological match demands in elite soccer players

**Andrea Riboli**[1,2]*, **Giuseppe Coratella**[2], **Susanna Rampichini**[2], **Emiliano Cé**[2], **Fabio Esposito**[2]

**1** Performance Department, Atalanta B.C., Bergamo, Italy, **2** Department of Biomedical Sciences for Health, Università degli Studi di Milano, Milano, Italy

* riboliandrea@outlook.com

**Data Availability Statement:** All relevant data are within the manuscript and its Supporting Information files.

## Abstract

The current study determined the area-per-player during small- or large-sided games with or without goalkeeper that replicates the relative (m·min⁻¹) total distance, high-intensity running distance, sprint distance and metabolic power covered during official matches. Time-motion analysis was performed on twenty-five elite soccer-players during 26 home-matches. A total of 2565 individual samples for SSGs using different pitch sizes and different number of players were collected and classified as SSGs with (SSG-G) or without goalkeeper (SSG-P). A between-position comparison was also performed. The area-per-player needed to replicate the official match demands was *largely* higher in SSG-G *vs* SSG-P for total distance [187±53 vs 115±35 m², effect size (ES): 1.60 95%CI 0.94/2.21], high-intensity running distance [262±72 vs 166±39 m², ES: 1.66(0.99/2.27)] and metabolic power [177±42 vs 94±40, ES: 1.99(1.31/2.67)], but similar for sprint distance [(316±75 vs 295±99 m², ES: 0.24(-0.32/0.79)] with direction of larger area-per-player for sprint distance > high-intensity running > total distance  metabolic power for both SSG-G and SSG-P. In SSG-G, forwards required higher area-per-player than central-defenders [ES: 2.96(1.07/4.35)], wide-midfielders [ES: 2.45(0.64/3.78)] and wide-defenders [ES: 3.45(1.13/4.99)]. Central-midfielders required higher area-per-player than central-defenders [ES: 1.69(0.20/2.90)] and wide-midfielders [ES: 1.35(-0.13/2.57)]. In SSG-P, central defenders need lower area-per-player (ES: -6.01/-0.92) to overall replicate the match demands compared to all other positions. The current results may be used to gain knowledge of the SSGs relative to the match demands. This imply manipulating SSGs using higher or lower ApP, the presence of the goalkeeper or design specific rules to increase or decrease the position-specific demands with respect to the desired external load outcomes.

**Funding:** The author(s) received no specific funding for this work

**Competing interests:** The authors have declared that no competing interests exist

## Introduction

Small- or large-sided games are frequently used to replicate the soccer-specific match demands in terms of technical proficiency, tactical awareness, speed, acceleration/deceleration, and endurance performance [1]. To assess these demands, contemporary player-tracking technologies such as global positioning system (GPS) or semi-automatic video-based multi-camera image system (MCIS), are typically used [2]. In small- or large-sided games (SSGs), the manipulation of pitch size, number of players per team, goalkeeper presence and technical rules modulate the soccer-specific demands depending on the aims of each practice session [1, 3]. Increments in pitch size or reduction in the number of players increases total distance (TD) covered, total high-intensity running distance (HIRD) and total sprint distance (TSD) [4, 5]. Conversely, when pitch size is reduced or the number of players is increased, players get more ball touches but they have not the space to reach the high-speed running, and the total distance covered is rather characterized by acceleration and deceleration (Acc/Dec) [5, 6]. To possibly combine the pitch size and number of players, the area per player (ApP, expressed as $m^2 \cdot player^{-1}$) has been introduced [1]. Lastly, SSGs can be performed with (SSG-G) or without goalkeepers (SSG-P), when the aim is to out-score the opponent team or to maintain ball possession as long as possible, respectively [1].Some authors reported higher TD and distances covered at different speed-thresholds during 2-, 3- and 4-a-side SSG-P than SSG-G [7]. Similarly, higher HIRD was found comparing 3-a-side [8], 5- and 7-a-side SSG-P than SSG-G [9]. Although TD, HIRD and TSD were found to be higher in SSG-P than SSG-G using the same pitch size [1, 10], other studies found lower HIRD in 3-a-side SSG-P than SSG-G [9], no differences in TSD in 3-a-side vs 5-a-side SSG-P than SSG-G [9] or higher TSD and lower Acc/Dec in SSG-G compared to SSG-P [6]. These conflicting findings suggest that further investigation is needed [11].

The metabolic power ($P_{met}$) approach has been recently proposed as a tool to estimate the energetic demands of variable-speed and accelerated/decelerated locomotion activities typically seen in team sports [12, 13]. While it is difficult to measure directly the exact energy cost of changing speed, a metabolic power calculation based on a theoretical model has been used to estimate the energy cost of locomotion in team sports [12, 14]. However, this model was questioned since it may underestimate the actual net energy demand of soccer-specific exercises [15–17]. Additionally, the traditional speed-threshold approach was shown to provide similar external load compared to $P_{met}$ [18, 19]. Nevertheless, the metabolic power approach could capture the high-demanding locomotor activities independently of the actual speed registered by GPS [16, 20], and it was shown to be a useful tool for the classification of the locomotion intensity in team sports [21]. Previous studies have provided evidence for concurrent ecological validity to this approach, reporting correlations between $P_{met}$ and aerobic fitness variables during professional soccer matches [22] and with time above 85% of the maximal heart-rate in elite hockey matches [21]. Moreover, $P_{met}$ can be sensitive to decrements in running performance during competition [23–25] and it could be used to account for positional differences [23, 25]. Therefore, the combination of the $P_{met}$ approach and the traditional speed-threshold metrics should be used to provides a more comprehensive assessment of the intermittent running demands typically occurring in team sports [15–17, 21, 24, 26–28].

An accurate comparison of the match *vs* training loads may help to plan the training sessions to condition the locomotor activities typically required during the official match and to optimize performance goals [5, 29]. Quantifying TD, HIRD, TSD, Acc/Dec and $P_{met}$ training loads relative to the game demands was suggested to be an important strategy when attempting to optimize position-specific loads in elite soccer practice [29]. Particularly, the locomotor activities during different SSGs compared to official matches are still under investigation.

Additionally, discriminating such locomotor activities by position could help to tailor the training session. Therefore, the present study aimed to: i) determine the ApP that could be used to replicate the official matches TD, HIRD, TSD, Acc/Dec (normalized as meters covered in one minute) and $P_{met}$ (normalized as $W \cdot kg^{-1}$) during both SSG-P and SSG-G; and ii) differentiate the ApP according to playing position. To increase the ecological validity, this was assessed in elite Serie A soccer players.

## Materials and methods

### Participants

Twenty-five elite soccer players competing in Italian Serie A were involved in the present study (age: 27 ± 5 yrs; body mass: 79 ± 7 kg; body height: 1.84 ± 0.06 m). All participants were classified according to their position: central-defenders (n = 6), wide-defenders (n = 4), central-midfielders (n = 5), wide-midfielders (n = 5) and forwards (n = 5). The goalkeepers were excluded from data collection. The club's medical staff certified the health status of each player. An injured player was excluded from data collection for at least one month after their return to full training. All participants gave their written consent after a full explanation of the purpose of the study and the experimental design. The Ethics Committee of the Università degli Studi di Milano approved the study and was performed in accordance with the principles of the Declaration of Helsinki (1975).

### Design

The present investigation was carried out during the competition period across two seasons (August 2014 –May 2016). The participants undertook their traditional weekly training routine. All sessions were performed on two grass pitches preserved by qualified operators and were conducted at the same time of day to limit the effects of circadian variation. A specialized and high-qualified physician staff recommended and monitored the diet regime of each player before and after every training session.

Two different formats of SSGs were analyzed: SSG-G and SSG-P. A total of 2565 (1033 and 1532, respectively) individual GPS samples with a median of 37 (range = 12 to 62) and 56 (range = 25 to 86) in SSG-G and SSG-P respectively were undertaken for each player. The number of players ranged from 5*vs*5 to 10*vs*10, with a pitch area ranging from 800 m$^2$ to 6825 m$^2$ for SSG-G and 3*v*3 to 10*vs*10 with a pitch area from 400 m$^2$ to 4550 m$^2$ for SSG-P. Hence, ApP ranged from 67 m$^2$ to 341 m$^2$ for SSG-G and from 43 m$^2$ to 341 m$^2$ for SSG-P (for a detailed description of these parameters, see S1 and S2 Tables). ApP was calculated excluding the goalkeepers in SSG-G. Both small- or large-sided games were abbreviated as SSGs and specified by ApP. The SSGs were performed under the supervision and motivation of several coaches to keep up a high work-rate [3]. For the same reason, a ball was always available by prompt replacement when it went out-of-play [1]. In SSG-G, the corners were replaced by a prompt ball-in-game from the goalkeeper [9]. The SSGs were completed after a standardized 20-min warm-up under the guidance of club staff. Only official home matches (N = 26; individual samples = 228; individual sample range = 6 to 24) were assessed to ensure data consistency [11]. The home-match pitch size was 105 x 66 m, with a grass surface.

To determine the ApP in both SSG-G and SSG-P that replicates the normalized TD, HIRD, TSD, Acc/Dec (m·min$^{-1}$) and $P_{met}$ (W·kg$^{-1}$) recorded during the official matches, we first recorded these variables during the official matches. Thereafter, we separately plotted each relationship between ApP and the normalized TD, HIRD, TSD, Acc/Dec and $P_{met}$ during SSG-G or SSG-P. Then, the mean values recorded during the official matches were used to

intersect each ApP/ TD, HIRD, TSD, Acc/Dec or $P_{met}$ relationship recorded in SSG-G or SSG-P to calculate the ApP that corresponded to the official match demands (Fig 1).

## Procedures

For the aims of this study, the interchangeability of GPS and MCIS for TD, HIRD, TSD, Acc/Dec and $P_{met}$ needed to be calculated as first step. A 10Hz GPS (K-Sport, Montelabbate, Italy) unit was used to collect data during the training sessions [30]. The GPS unit was placed within a dedicated pouch between the player's shoulder blades (upper thoracic-spine) in a sports vest and worn under the playing jersey. Each device was turned on at least 15-min before each session to allow for acquisition of the satellite signal [6]. To reduce the inter-unit differences, each player wore the same unit for every training session over the whole investigation [31]. The locomotor activities during the official matches were collected using a computerized semi-automated MCIS (STATS LLC, Chicago, Illinois, USA) and processed by a dedicated software (K-SportOnline, K-Sport, Montelabbate, Italy). The system has previously been shown to provide valid and reliable measurements of the match activity in soccer [32, 33].

During both training sessions and home-matches, total distance, total high-intensity running distance ($>15$ km·h$^{-1}$), total sprint distance ($>24$ km·h$^{-1}$) [3, 11, 33] were measured.

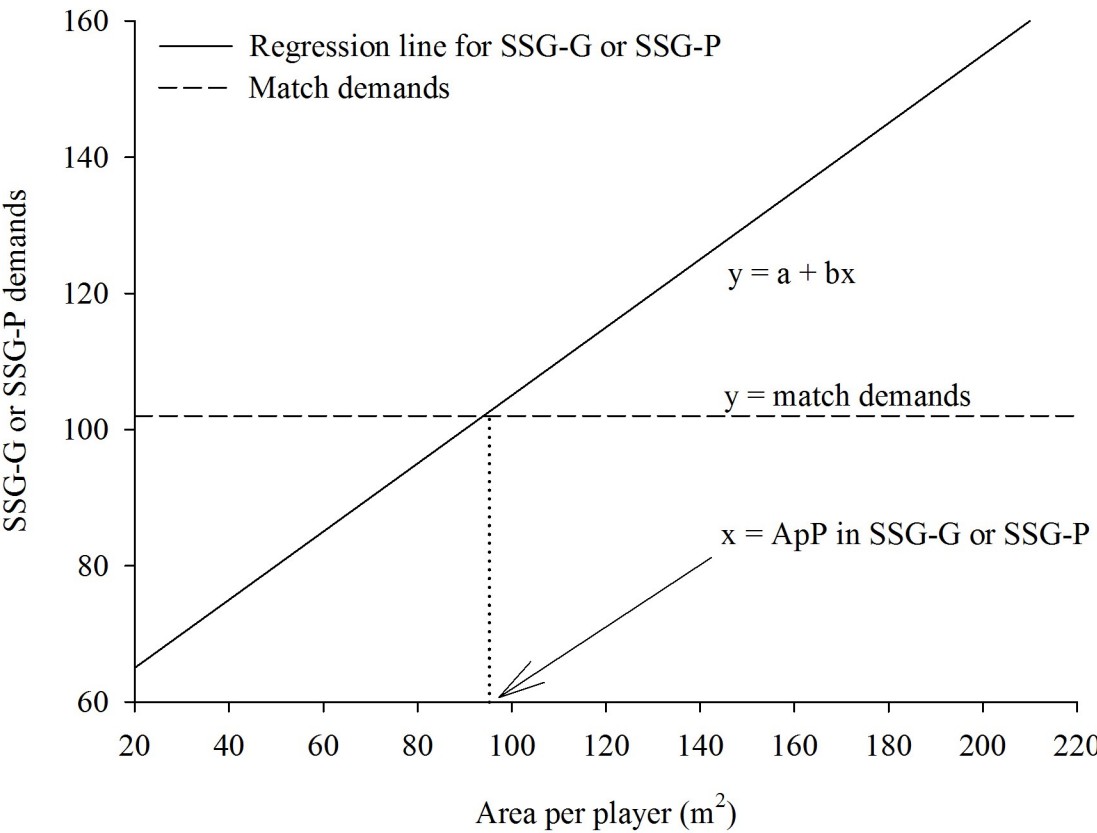

**Fig 1. Graphical representation of the procedures used to determine the area per player in SSG-G or SSG-P that matches the official match demands.** X-axis: the area per player in SSG-G or SSG-P; Y-axis: the SSG-G or SSG-P demands. The regression line shows how the area per player influences the SSGs demands. The horizontal dashed line represents the official match demands. From the intersection point of the regression line with the horizontal line (i.e. when the SSGs demands equate the official match demands), a vertical dotted line is drawn to the X-axis. The intersection point between the X-axis and the vertical dotted line is the calculated area per player in SSGs necessary to replicate the official match demands.

Additionally, the total distance of velocity changes calculated using $>2$ m·s$^{-2}$ accelerations and decelerations (Acc/Dec) were measured [4, 5]. The average metabolic power ($P_{met}$) was calculated following previous procedures [13, 27]. TD, HIRD, TSD and Acc/Dec were normalized as relative distance covered in one minute (m·min$^{-1}$), while $P_{met}$ were normalized as watt per kilogram (W·kg$^{-1}$); then all parameters were inserted into the data analysis.

TD, HIRD, TSD and Acc/Dec were measured using either GPS or the MCIS. Therefore, to check the interchangeability of these two tracking technologies, a 10-min simulated match was monitored using both GPS and MCIS simultaneously [2, 34, 35]. All data were collected in the stadium where the official matches were played. For each dependent locomotor activity, a calibration equation was calculated to compare GPS and MCIS, as previously proposed [2, 34].

### Statistical analysis

Statistical analysis was performed using a statistical software package (SigmaPlot v-12.5, Systat Software Inc., San Jose, CA, USA). To check the normal distribution of the sampling, a Shapiro-Wilk test was used. A Bland-Altman analysis was used to display the degrees of bias and the limits of agreement between the GPS and the MCIS. A linear regression analysis was used to calculate the correlation between GPS and MCIS. The Pearson's product moment and the typical error of the estimate (TEE) were calculated to determine the relationship between the GPS and the MCIS. The correlation coefficient was interpreted as follows: $r$ = 0.00–0.09 *trivial*, 0.10–0.29 *small*, 0.30–0.49 *moderate*, 0.50–0.69 *large*, 0.70–0.89 *very large*, 0.90–0.99 *nearly perfect*; the threshold values for the TEE were interpreted as follows: >0.2 *small*, >0.6 *moderate*, >1.2 *large* and >2 *very large* [36]. A linear regression analysis was used to calculate the correlation between TD, HIRD, TSD, Acc/Dec, $P_{met}$ and the ApP during both SSG-G and SSG-P. Thereafter, a two-way ANOVA was used to calculate the difference in the optimal ApP in TD, HIRD, TSD, Acc/Dec, $P_{met}$ calculated for SSG (SSG-G *vs* SSG-P) and position (central-defenders, wide-defenders, central-midfielders, wide-midfielders and forwards). A post-hoc analysis (Holm-Sidak correction) was used to calculate the differences in the independent factors. The effect size with 95% confidence intervals (CI) was calculated and interpreted as follows: <0.20: *trivial*; 0.20–0.59: *small*; 0.60–1.19: *moderate*; 1.20–1.99: *large*; ≥2.00: *very large* [36]. Statistical significance was set at $\alpha < 0.05$. Unless otherwise stated, all values are presented as mean ± standard deviation (SD).

### Results

The magnitude of the GPS *vs* MCIS bias is shown in Fig 2. For each dependent parameter, Bland-Altman analysis and correlation graph with the respective calibration equation are shown. The bias between GPS *vs* MCIS were *trivial* for TD (-3.0 ± 1.3%, ES = -0.18, CI: -0.80/0.44), HIRD (-3.3 ± 1.6%, ES = -0.12, CI: -0.74/0.51), TSD (-3.9 ± 10.9%, ES = -0.11, CI: -0.44/0.22) and Acc/Dec (-4.1 ± 6.3%, ES = -0.19, CI: -0.80/0.44) and *small* for $P_{met}$ (-4.0 ± 0.6%, ES = -0.38, CI: -1.27/0.49). A *small* TEE was found between the MCIS and GPS for TD (TEE: 0.09, CI: 0.07/0.14), HIRD (TEE: 0.04, CI: 0.03/0.06), TSD (TEE: 0.08, CI: 0.07/0.10) and $P_{met}$ (TEE: 0.07, CI: 0.04/0.13), while a *moderate* TEE was found for Acc/Dec (TEE: 0.75, CI: 0.56/1.10). In addition, a *nearly perfect* correlation was observed for TD, HIRD, TSD and $P_{met}$ and a *moderate* correlation for Acc/Dec measured using GPS and MCIS (Fig 2).

As shown in Fig 3, in SSG-P a *very large* correlation between the relative distance and the ApP was found for TD, HIRD, TSD and $P_{met}$ while a *moderate* correlation was found for Acc/Dec. In SSG-G, a *very large* correlation between the relative distance and the ApP for TD, HIRD and TSD, a *large* correlation for $P_{met}$ and a *moderate* negative correlation for Acc/Dec

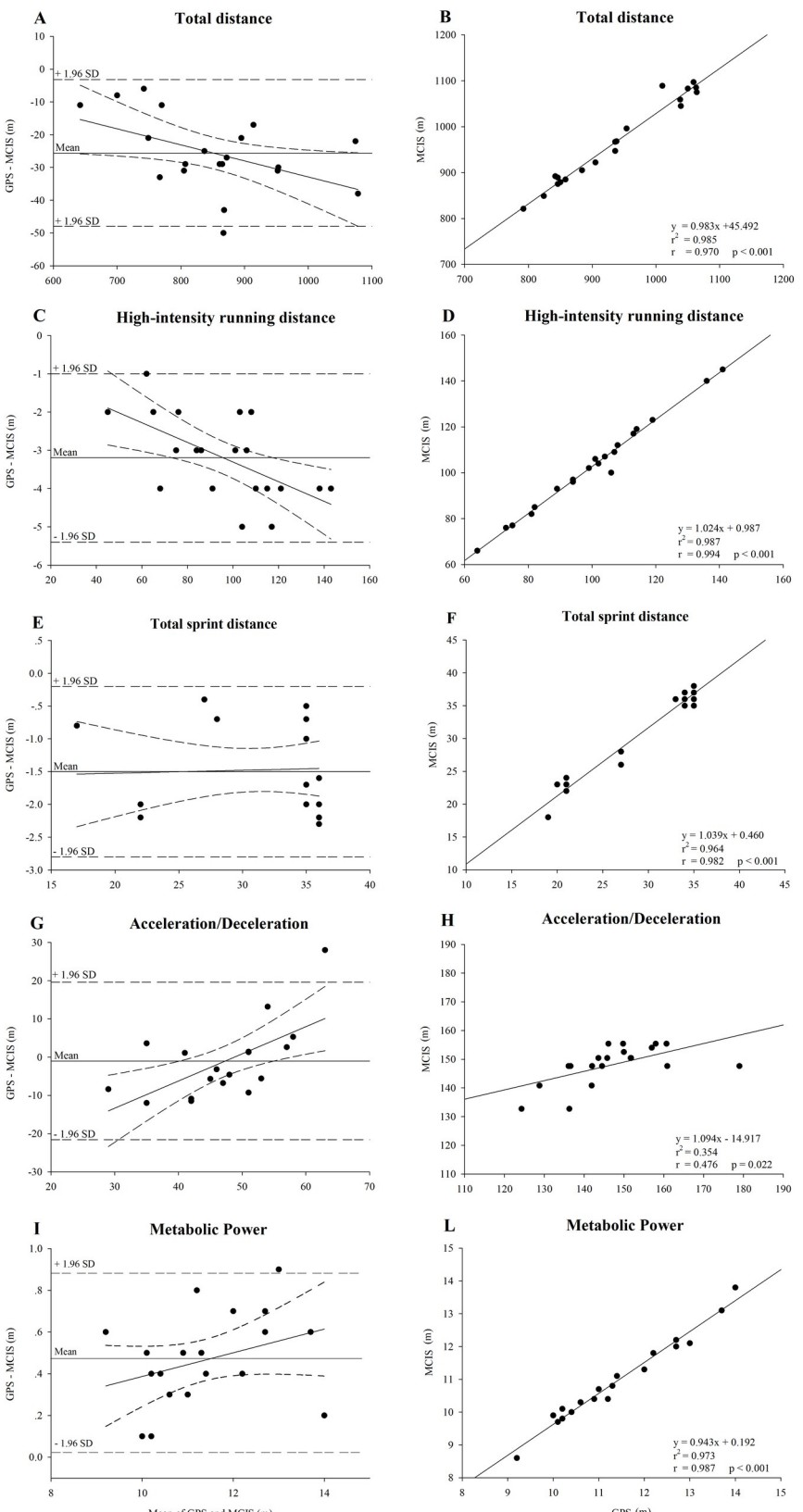

**Fig 2. Bland-Altman analysis and linear regression analysis with calibration equation for the GPS *vs* MCIS bias for each locomotor activity.** The linear regression analysis is shown with 95% confidence bands. Panels A-B: total distance; C-D: high-intensity running distance; E-F: total sprint distance; G-H: acceleration/deceleration; I-L: metabolic power.

were found. Because of the *moderate* correlations observed for Acc/Dec in both SSG-G and SSG-P, we did not perform the calculation or the ApP for Acc/Dec, given the high risk of bias.

For both SSG-P and SSG-G, the ApP necessary to replicate the relative distance recorded during the matches for TD, HIRD, TSD and $P_{met}$ is shown in Table 1. No SSG × position interaction was found (p = 0.674) for ApP for TD. A main effect for SSG (p < 0.001) and position (p = 0.024) was detected. The between-SSG post-hoc analysis is reported in Table 1. In SSG-P, a larger ApP is required for forwards *vs* central-defenders (p = 0.023; ES = 4.35, CI: 1.93/6.01), with no other between-position differences. In SSG-G, no between-position difference occurred.

No SSG × position interaction was found (p = 0.065) for ApP for HIRD. A main effect for SSG (p < 0.001) and position (p < 0.001) was detected. The between-exercise post-hoc analysis is reported in Table 1. In SSG-P, a higher ApP is required for forwards *vs* central-defenders (p = 0.024; ES = 2.92, CI: 1.04/4.29), with no other between-position differences. In SSG-G, forwards required higher ApP than central-defenders (p < 0.001; ES = 2.96, CI: 1.07/4.35), wide-midfielders (p = 0.002; ES = 2.45, CI: 0.64/3.78) and wide-defenders (p = 0.029, ES = 3.45, CI: 1.13/4.99). Central-midfielders required a higher ApP than central-defenders (p = 0.002; ES = 1.69, CI: 0.20/2.90) and wide-midfielders (p = 0.019, ES = 1.35, CI: 0.13/2.57).

No SSG × position interaction was found (p = 0.803) for ApP for TSD, not even a main effect for exercise (p = 0.415). A main effect for position (p = 0.049) was detected. The between-exercise post-hoc analysis is reported in Table 1. In both SSG-P and SSG-G, no between-position difference occurred.

No SSG × position interaction was found (p = 0.167) for ApP for $P_{met}$. A main effect for SSG (p < 0.001) and position (p = 0.002) was detected. The between-SSG post-hoc analysis is reported in Table 1. In SSG-P, a lower ApP is required for central-defenders *vs* wide-defenders (p = 0.031; ES = -2.69, CI: -4.32/-1.05), wide-midfielders (p < 0.001; ES = -2.64, CI: -4.35/-0.92), central-midfielders (p = 0.028; ES = -5.10, CI: -7.53/-2.66), forwards (p = 0.024; ES = -1.89, CI: -3.32/0.47). In SSG-G, no between-position difference occurred.

## Discussion

The first novel finding observed in the present study was a detailed calculation of the ApP in SSG-P or SSG-G necessary to replicate the TD, HIRD, TSD or $P_{met}$ recorded during the official matches. It is shown here that, irrespective of the SSG type, the higher the speed threshold, the larger the ApP required (i.e., TSD > HIRD > TD ≈ $P_{met}$). Secondly, the inclusion of the goal-keeper increases the ApP for TD, HIRD and $P_{met}$, while no difference was observed in SSG-P *vs* SSG-G for TSD. Additionally, central defenders required the lowest ApP compared to all other positions, both in SSG-P and SSG-G. Lastly, both central-midfielders and forwards need the highest ApP compared to all other positions, both in SSG-G and SSG-P, to replicate the match demands.

During official matches, total high-intensity running distance covered [37], technical skills to maintain greater ball possession [38], the total distance covered with ball possession [39] and tactical behaviours [40] are key factors for success in soccer performance. Within weekly training routines, SSGs are largely used to elicit high-intensity running [1], a high number of technical drills with the ball possession [41] and to improve tactical behaviours [40].

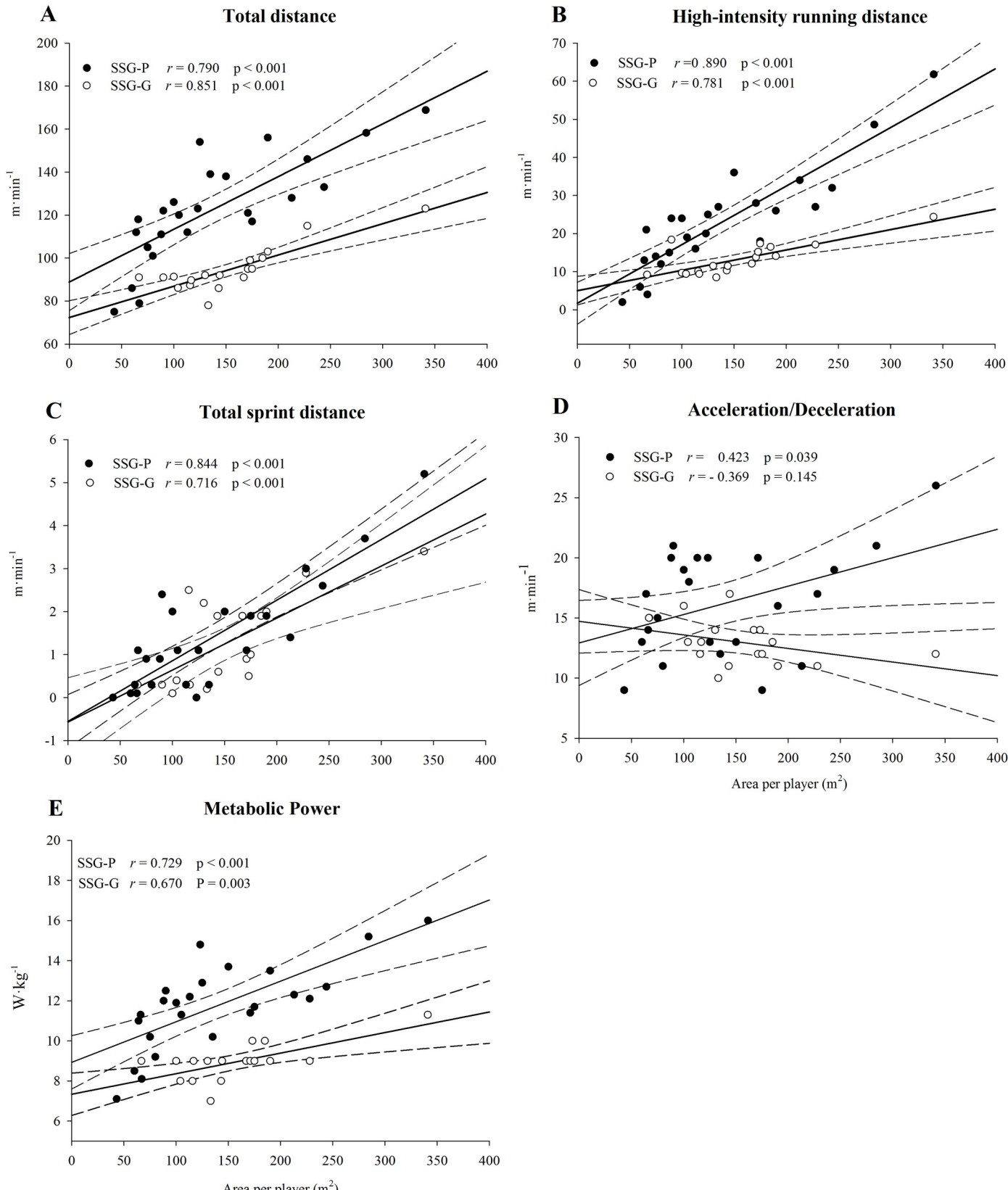

**Fig 3. The relationship between area per player (m$^2$·player) and relative speed distance (m·min$^{-1}$) or estimated metabolic power (W·kg$^{-1}$) for each locomotor activity.** The linear regression analysis with 95% confidence bands and the correlation between the area per player and the relative distance or metabolic power are also reported. SSG-P, closed circles: small-sided games possession-play without goalkeepers; SSG-G, open circles: small-sided games with goalkeepers. Panel A: total distance; B: high-intensity running distance; C: total sprint distance, D: acceleration/deceleration; E: metabolic power.

Interestingly, SSGs were shown to lead to similar enhancement in aerobic fitness than high-intensity running interval training [42]. In SSGs, manipulating the number of players, the pitch size and the goalkeeper presence results in different physiological, technical and tactical outcomes [1, 40]. For example, while increments in ApP was shown to increases TD, HIRD and TSD [4, 5], decreasing ApP leads to more ball touches and Acc/Dec [5, 6]. Determining the ApP that replicates the match external-load demands may help sport physiologists and practitioners to properly plan SSGs for specific performance objectives [5]. Therefore, the current results may be used to gain knowledge of the SSGs relative to the match demands. Unsurprisingly, both in SSG-P and in SSG-G, higher ApP leads to greater distance covered whatever the speed threshold [5]. Accordingly, the present findings highlight that the ApP in SSGs to replicate the TSD match demands is very close to the official match ApP ($\approx 340$ m$^2$). In line with the present outcomes, it was shown that the larger the pitch size, the greater the distance covered at speed >18 km·h$^{-1}$ [43]. Other authors found that TD and the distance covered at 19.8–25.2 km·h$^{-1}$ and >25.2 km·h$^{-1}$ increased proportionally with the pitch size [44]. A recent study reported that ApP $\approx 311$ m$^2$ was able to replicate the high-speed match demands during SSG-G [5]. The exposure to high-demanding activities was shown to improve the players'

**Table 1. Area per player (m$^2$·player) to replicate official-match load using SSGs for relative speed distances or estimated metabolic power.** Data are presented as mean(SD), effect size (95% confidence intervals).

| Position | TD | | | | HIRD | | | | TSD | | | | $P_{met}$ | | | |
|---|---|---|---|---|---|---|---|---|---|---|---|---|---|---|---|---|
| | SSG-P | SSG-G | p | ES (CI) | SSG-P | SSG-G | p | ES (CI) | SSG-P | SSG-G | p | ES (CI) | SSG-P | SSG-G | p | ES (CI) |
| **Total** | 115 (35) | 187 (53)[a] | <0.001 | -1.60 (-2.21/-0.94) | 166 (39) | 262 (72)[a] | <0.001 | -1.66 (-2.27/-0.99) | 295 (99) | 316 (75) | 0.415 | -0.24 (-0.79/0.32) | 94(40) | 177 (42)[a] | <0.001 | -1.99 (-2.67/-1.31) |
| **CD** | 65 (24)[b] | 165 (26)[a] | <0.001 | -4.00 (-5.55/-1.83) | 122 (30)[b] | 205 (57)[abc] | <0.001 | -1.82 (-3.00/-0.37) | 257 (76) | 278 (51) | 0.672 | -0.32 (-1.44/0.84) | 31(11) | 151 (23)[a] | <0.001 | -6.14 (-8.85/-3.44) |
| **WD** | 121 (21) | 193 (71)[a] | 0.023 | -1.38 (-2.70/0.31) | 163 (30) | 246 (36)[ab] | 0.003 | -2.50 (-3.92/-0.43) | 297 (26) | 274 (67) | 0.696 | 0.45 (-1.01/1.79) | 106 (31)[d] | 183 (27)[a] | 0.003 | -2.39 (-4.01/-0.77) |
| **CM** | 119(9) | 184 (41)[a] | 0.021 | -2.12 (-3.41/-0.42) | 174 (28) | 311 (69)[a] | <0.001 | -2.60 (-3.96/-0.74) | 329 (66) | 340 (33) | 0.834 | -0.21 (-1.43/1.05) | 107 (13)[d] | 191 (25)[a] | <0.001 | -3.81 (-5.88/-1.73) |
| **WM** | 135 (20) | 183 (81)[a] | 0.079 | -0.81 (-2.02/0.55) | 172 (19) | 222 (72)[abc] | 0.031 | -0.95 (-2.15/0.44) | 264 (52) | 281 (62) | 0.758 | -0.30 (-1.51/0.98) | 132 (12)[d] | 180 (61)[a] | 0.047 | -0.95 (-2.71/0.51) |
| **FW** | 147(9) | 214 (51)[a] | 0.018 | -1.83 (-3.09/-0.22) | 207 (28) | 333 (12)[a] | <0.001 | -5.85 (-7.91/-2.66) | 334 (92) | 407 (68) | 0.176 | -0.90 (-2.10/0.48) | 99(2)[d] | 201 (66)[a] | <0.001 | -1.97 (-3.48/-0.46) |

TD, total distance; HIRD, high intensity running distance; TSD, sprint distance; $P_{met}$, average metabolic power; SSG-P, small-sided games without goalkeepers; SSG-G, small-side games with goalkeepers; Total, team average; CD, central defenders; WD, wide defenders; CM, central midfielders; WM, wide midfielders; FW, forwards; ES, effect size; CI, confidence interval.

[a] Significantly different (p < 0.05) from SSG-P

[b] Significantly different (p < 0.05) from forwards

[c] Significantly different (p < 0.05) from central midfielders

[d] Significantly different (p < 0.05) from central defenders

fitness level, to prepare the players to the match workload and to result in greater protection against non-contact injuries [45–47]. Therefore, manipulating ApP allows for training loads in SSGs to be managed with respect to the desired external load outcomes, both for performance and prevention purposes.

The current findings also highlight that training using SSGs with or without goalkeeper affects the ApP necessary to replicate the match demands. Particularly, with the exception of TSD, the goalkeeper presence increases the ApP for TD, HIRD and $P_{met}$, i.e. SSG-G > SSG-P. Partially in contrast with the present outcomes, it was reported that SSG-G resulted in higher TSD than found in SSG-P [6]. However, the authors investigated a maximum ApP of 135 m$^2$, hence, this does not allow an appropriate comparison. Other researchers reported that TD and the time spent in high-intensity running (>17 km·h$^{-1}$) was higher with goalkeepers [48]. Although the authors argued that the goalkeeper presence might have motivated the players, several authors found higher high-intensity running without goalkeepers in different 3-, 4-, 5-, and 7-a-side SSGs [7–9]. Moreover, two subsequent reviews [1, 10] consistently remarked that the goalkeeper presence could improve the players' organization, thus decreasing the SSGs demands. Indeed, during SSG-G, the two teams' aim is to outscore the opponent team, while maintaining a match-like tactical organization. In contrast, since during SSG-P the aim is to maintain the ball possession as long as possible, the players are free to move across the selected pitch size. This rule-difference seems to account for the largest ApP in SSG-G necessary to replicate the TD, HIRD or $P_{met}$ recorded during the official matches. Interestingly, the current results come with *moderate* correlation between Acc/Dec and ApP in SSG-P, while no correlation was observed between Acc/Dec and ApP in SSG-G. Previous results suggested that lower pitch size induced increments in Acc/Dec [5, 49]. In line with the present outcomes, other authors retrieved no differences for high-demand (>2 m·s$^2$) Acc/Dec with the increment in pitch size during 3-, 5- and 7-a-side [9] or 3-, 5-, and 10-a-side SSG-G or SSG-P [6]. Comparing SSG-P vs SSG-G, higher Acc/Dec were reported during SSG-G than SSG-P using an ApP of ~210 m$^2$ [9], while no difference in Acc/Dec between SSG-G vs SSG-P were found using an ApP from 73-to-135 m$^2$ [6]. Despite the greater stimulation of accelerations in SSG-G vs SSG-P might be accounted for the players' need to overpass the opponent or defensive lines in order to achieve the rival goal in larger ApP, a controversy still exists.

To our knowledge, the calculation of the ApP across positions was used here for the first time. No between-position difference in ApP was found for TSD, neither in SSG-P nor SSG-G. In SSG-P, it was observed that central defenders need lower ApP than forwards for TD, and HIRD, while lower ApP than all other position for $P_{met}$. In SSG-G, no between-position difference in ApP was observed for TD, TSD and $P_{met}$, while forwards and central midfielders need larger ApP than central defenders and wide midfielders for HIRD suggesting that these positions might undergo different stimuli during similar SSGs. Defenders tend to move within a "defined" space over the official match, while central-midfielders and forwards tend to cover a greater area of the pitch in order to gain possession of the ball, marking the opponent or creating space to score [33]. This might be considered for the lower ApP needed to accumulate the match demands in central defenders than forwards/central-midfielders. However, the sprinting activities are not influenced by position, since these appear to need large pitch areas available anyhow. The different ApP recorded across position offers the possibility to tailor the training load to enhance the performance adaptations. It was previously suggested that similar high-intensity training load could lead to overload or underload different positions, so affecting the competition performance or possibly increasing the risk of injury [29]. Interestingly, high-intensity activities were shown to be underloaded during the training routines compared to the official matches, with a high variability across positions [29]. The present results suggest that some positions need higher or lower ApP to replicate the HIRD or TSD accumulated over

the matches. Furthermore, position-specific rule modifications within SSGs or additional exercises could be integrated to technical/tactical exercises to individualize high-intensity training activities.

Some limitations accompany the present investigation. For replication purposes, the interchangeability between the GPS and MCIS needs to be carefully checked, especially when recording high-speed or non-linear movements [2]. The present results are based on the *trivial* differences in the metrics recorded using either the GPS or MCIS and a calibration equation was provided to partially account for these differences. Secondly, due to technological limitation during the official matches, no internal load parameter (e.g. heart rate) was assessed. However, it was reported that $P_{met}$ maintains a strong and consistent relationship with the measures of internal load during low-to-high intensity locomotor activities [21]. Therefore, $P_{met}$ could be a satisfactory way to estimate with accuracy the training and match demands [12, 22] and to classify the locomotion intensity in team sports [21, 28].

## Conclusions

The current results suggest that soccer players need a specific ApP during SSGs with or without goalkeeper to replicate the match demands, especially to perform each locomotor activity (i.e., TSD > HIRD > TD ≈ $P_{met}$). Moreover, SSG-G need higher ApP than SSG-P to replicate the match demands. Lastly, position-difference in ApP were found, so that central defenders need lower and forwards and central midfielders higher ApP.

These results allow managing the training loads towards the desired players' fitness component to maximize transfer to the game-like and performance goal using SSGs. Indeed, soccer training methodology are evolving to an alternation of the training objectives with the aim to overload the desired fitness component relative to the match demands [5, 29]. When aware of the training/matches differences in locomotor activities, coaches could design SSGs with the intent to replicate, underload or overload the match demands. This imply manipulating SSGs using higher or lower ApP, the presence of the goalkeeper or design specific rules to increase or decrease the position-specific demands. To synthetize, the present outcomes could be used in practice to: i) calculate an ApP that replicate an estimated match demand using $P_{met}$ for both SSG-P and SSG-G; ii) replicate the official relative match demands using the specific minimal ApP to HIRD or TSD be accumulated during the SSG-G/P performed in the training sessions; iii) differentiate the ApP when SSG-P or SSG-G are performed according to the aim of the training session (e.g. replicate, overload or underload specific training objectives); iv) add SSGs with position-specific ApP to the training routines when needed or propose specific additional exercises or rules to overload or underload each player.

## Supporting information

**S1 Table. Small-sided games with goalkeepers.** The small-sided games with goalkeepers are split for the number of players and pitch size (width x length). The total pitch area and area per player have been calculated. The average number of observations per player for each condition are also reported as mean (max-min).
(DOCX)

**S2 Table. Small-sided games without goalkeepers.** The small-sided games without goalkeepers are split for the number of players and pitch size (width x length). The total pitch area and area per player have been calculated. The average number of observations per player for each condition are also reported as mean (max-min).
(DOCX)

**S1 Data.**
(XLSX)

## Author Contributions

**Conceptualization:** Andrea Riboli.

**Data curation:** Andrea Riboli.

**Formal analysis:** Emiliano Cé.

**Investigation:** Andrea Riboli.

**Methodology:** Andrea Riboli, Giuseppe Coratella.

**Software:** Susanna Rampichini.

**Supervision:** Fabio Esposito.

**Visualization:** Giuseppe Coratella.

**Writing – original draft:** Andrea Riboli, Giuseppe Coratella.

**Writing – review & editing:** Andrea Riboli, Giuseppe Coratella, Fabio Esposito.

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
