## [Decision Letter · Decision Letter 0]

25 Mar 2020

PONE-D-20-02888

Area per player in small-sided games to replicate the external load and estimated physiological match demands in elite soccer players

PLOS ONE

Dear Dr. RIBOLI,

Thank you for submitting your manuscript to PLOS ONE. After careful consideration, we feel that it has merit but does not fully meet PLOS ONE’s publication criteria as it currently stands. Therefore, we invite you to submit a revised version of the manuscript that addresses the points raised during the review process.

Please, address point-to-point all reviewers' issues, in particular Reviewer 3's ones. In its current format MS is not suitable for publication.

We would appreciate receiving your revised manuscript by May 09 2020 11:59PM. To enhance the reproducibility of your results, we recommend that if applicable you deposit your laboratory protocols in protocols.io, where a protocol can be assigned its own identifier (DOI) such that it can be cited independently in the future. For instructions see: http://journals.plos.org/plosone/s/submission-guidelines#loc-laboratory-protocols

We look forward to receiving your revised manuscript.

Kind regards,

Luca Paolo Ardigò, Ph.D.

Academic Editor

PLOS ONE

Journal Requirements:

2. Thank you for inclkuding your ethics statement; "All participants gave their written consent after a full explanation of the purpose of the study and the experimental design. The local University Ethics Committee approved the study and was performed in accordance with the principles of the Declaration of Helsinki (1975). "

Additional Editor Comments (if provided):

Please, address point-to-point all reviewers' issues, in particular Reviewer 3's ones. In its current format MS is not suitable for publication.

Reviewers' comments:

Reviewer's Responses to Questions

**Comments to the Author**

1. Is the manuscript technically sound, and do the data support the conclusions?

Reviewer #1: Yes

Reviewer #2: Yes

Reviewer #3: Partly

2. Has the statistical analysis been performed appropriately and rigorously? 

Reviewer #1: Yes

Reviewer #2: Yes

Reviewer #3: Yes

3. Have the authors made all data underlying the findings in their manuscript fully available?

Reviewer #1: Yes

Reviewer #2: Yes

Reviewer #3: Yes

4. Is the manuscript presented in an intelligible fashion and written in standard English?

Reviewer #1: Yes

Reviewer #2: Yes

Reviewer #3: Yes

5. Review Comments to the Author

Reviewer #1: I congratulate the authors for the excellent and innovative work presented. This represents a huge effort and dedication compensated with outstanding results for better knowledge and application of SSG

Reviewer #2: The purpose of this study was to determine the area-per-player during small- or large-sided games with or without goalkeeper that replicates the relative (m�min-1) total distance, high-intensity running distance, sprint distance and metabolic power covered during official matches. The paper is very interesting, nevertheless, I would like to suggest some ideas in order to clarify the reading of the text. I hope with my comments the quality of the study can be improved.

Specific comments:

Introduction

The main idea that is discussed in the introduction is related to compare physical demands between small-sided games played with and without goalkeepers. In this line, I believe you need to include all references that study this comparison. Here some of them:

• Sassi, Reilly, & Impellizzeri (2004)

• Mallo, J., y Navarro, E. (2008). Physical load imposed on soccer players during small-sided training games. Journal of Sports and Physical Fitness, 48(2), 166-171.

• Casamichana, D., Castellano, J., González-Morán, A., García-Cueto, H., y García-López, J. (2011). Demanda fisiológica en juegos reducidos de fútbol con diferente orientación del espacio. Revista Internacional de Ciencias del Deporte, 23(7), 141-154.

• Köklü Y, Sert O, Alemdaroğlu U, & Arslan Y. (2015). Comparison of the physiological responses and time motion characteristics of young soccer players in small sided games: the effect of goalkeeper. Journal of Strength and Conditioning Research,

• Joaquín González-Rodenas1, Ferran Calabuig1, Rafael Aranda1. Effect of the Game Design, the Goal Type and the Number of Players on Intensity of Play in Small-Sided Soccer Games in Youth Elite Players. Journal of Human Kinetics volume 49/2015, 229-235

Methods

Procedure

Please clarify (maybe including interchangeability formulas) more information about interchangeability.

Line 166, than or then?

Line 169 please specify what the system used to register each variable

Lines 172-173, about calibration equations, were they calculated or copied from Buccheit’s study. Please clarify.

Statistical analysis

Could you include Bland and Altman plot statistics to determine the relationship between the GPS and the MCIS?

Results

I have tried to put the values appear in the table 1 in the figures A, B, C and D, but they do not match in the regression lines, am I doing something wrong? (e.g., the total value for all position of 115 m in SSG-P or 187 in SSG-G added to the figure A do not match with the reference line of the game and regression lines). Please explain.

Discussion

Please remove all the sections inside this section (e.g., Methodological consideration”, “main findings”…)

All information about methodological consideration must include like limitations of the study, in the last part of the discussion section. Other limitation that the authors must considerer include could be that no reference about number of players per team used in the tasks. It is not the same to play a 4:4 in a relative space of 100 m per player, that one 8:8 in the same relative space per player.

Line 330, freer?¿?¿?

Please add to the discussion section the connection with the papers suggested in the introduction.

The idea argued in this paragraph (from line 318 to 332) could be improved adding information that from my point of view is important. Because previous academic literature (Castellano et al., 2013) suggest the variables regarding to acceleration are more demanded in oriented games. So depending of the variable considered the task could be more or less demanded. The idea behind this more stimulation of accelerations in task with goalkeepers is that players need to overpass opponent or defensive lines in order to achieve the rival goal. Maybe could be interesting to include this information and discuss about it.

The last section, conclusion, I think could be interesting how the coaches could apply the result suggested in this study. Different position needs no same ApP, but how can coaches to design game tasks following the results. I think include this applicability is key.

Reviewer #3: General Comments

I think the basic premise of this study is sound, and the data potentially provides meaningful and interesting information for sport science researchers and practitioners. However, I feel there are a few important issues that the Authors need to consider in revising this manuscript.

Firstly, the whole premise of that study is to compare various SSG formats to match conditions, and yet the measurement of match conditions is with an entirely different measurement system. It is not sufficient to apply an existing “calibration equation”, as this was developed from entirely different systems. If the Authors want to compare data between their current systems, they need to, at the very least, present some pilot data that compares the output from their own GPS & MCIS products.

Secondly, the Authors seem to rely on 2 articles which question the validity of the metabolic power model whilst ignoring subsequent research which supports its application in team sports (and addresses some of the misconceptions and flaws of those early papers. Meanwhile, their own results seem to indicate that metabolic power is an appropriate tool for monitoring the demands of SSGs, but the Authors appear to overlook this finding.

Thirdly, the Authors seem to focus too much on whether a particular SSG replicates the match demands, and neglect to adequately address some of the other objectives and advantages of SSGs. At a very simplistic level, if the intent is merely to replicate the demands/characteristics of a match, then as a reader I find myself asking “why not just play matches then?”. The Authors need to provide more context and perspective regarding the role of SSGs in training.

Specific Comments

Introduction

Page 3 Line 52 – I suggest this sentence be re-worded to read “To assess these demands, contemporary player-tracking technologies such as global positioning system (GPS) or semi-automatic video-based multi-camera image systems (MCIS), are typically used.” Note that “system” should be singular.

Page 3 Line 61 – there are numerous issues with the term “MechW” mentioned here. Firstly, when the authors state that acceleration and deceleration is “often referred to as mechanical work”, the paper that they cite (Gaudino 2014) never uses the term nor the acronym. Secondly, and more importantly, by defining that MechW applies to changes in speed, this implies that no mechanical work is done when running at a constant speed. This is clearly not the case. The Authors need to abandon the term MechW throughout the manuscript and adopt more appropriate terminology. Just as importantly, this needs to filter through to the way the Authors interpret and discuss their findings.

Page 3 Line 66 – change “reviewed” to “found”.

Page 3 Line 69 – change “Over the last years” to “In recent times”.

Page 4 Line 75 – the statement “it does not take into account the individual running economy” is a throw-away line that lacks evidential support. Neither of the papers cited here actually assessed individual economy, and in any case it is the COST rather than ECONOMY that is used to estimate metabolic power. Meanwhile, there is substantial experimental evidence in the literature that the cost of locomotion varies little between individuals.

Page 4 Line 77 – the statement here indicates a misunderstanding of the metabolic power model, as it implies that it is only relevant for assessing accelerations and decelerations but not high-speed running. This represents one of many misconceptions from previous literature that the Authors are perpetuating.

Material and Methods

Page 5 Line 98 – remove “the” prior to “data collection”.

Page 5 Line 99 – remove “the” prior to “data collection”.

Page 5 Line 100 – change “…full in-group re-integration.” to “…their return to full training.”

Page 5 Line 106 – change “…during the in-season period (August 2014 – May 2016)” to “during the competition period across two seasons (August 2014 – May 2016)”.

Page 5 Line 108 – change “All sessions were performed on two grass pitches preserved by qualified operators and performed at the same time to limit the effects of circadian variations” to “All sessions were performed on two grass pitches preserved by qualified operators and were conducted at the same time of day to limit the effects of circadian variation”. Note here that “variation” is singular.

Page 5 Line 118 – change “S1 and S2 Tables” to “Tables S1 and S2”.

Page 5 Line 119 – change “Both small- or large-sided games was abbreviated…” to “Both small- and large-sided games were abbreviated…”

Page 6 Line 124 – change “…warm-up under the club’s staff guidance” to “warm-up under the guidance of club staff”.

Page 7 Line 153 – change “each data collection” to “each session”.

Page 7 Line 153 – change “allow the acquisition” to “allow for acquisition”.

Page 7 Line 154 – change “every player wore the same device for each training session” to “each player wore the same unit for every training session”. Note that “unit” should be used instead of “device”.

Page 8 Line 173 – the “calibration equations” proposed by Buchheit et al. are specific to the devices and tracking systems assessed in that study. There is no reason to assume that they would equally apply to the GPS and MCIS brands used in the present study, and therefore makes the use of this equation dubious.

Discussion

General Comments – in addition to determining what ApP values replicate match demands for each position, I think it is important that the Authors acknowledge (and discuss) that SSGs also have a role in overloading different aspects of physical demands as well as emphasising various tactical or technical parameters. If the only objective of SSGs was to replicate match demands, then surely teams would just play matches. I think the Authors need to include this point in their interpretation and discussion of the results.

Page 13 Line 273 – this first sentence is a bit clumsy with respect to wording. I suggest something like “In order to properly interpret the results observed in the present study, some preliminary considerations need to be taken into account”.

Page 13 Line 281 – change “amount” to “magnitude”.

Page 13 Line 284 – again here the Authors are perpetuating a misconception surrounding the metabolic power model, and ignoring more recent and relevant literature which demonstrates that the model is quite accurate in estimating the energetic cost of LOCOMOTION for variable-speed activity (e.g. Osgnach & di Prampero, 2018)

Page 13 Lines 286-288 – I think this is an example of where the Authors rely too much on one or two papers rather than providing a more extensive and balanced assessment of existing literature.

Page 13 Line 293 – change “here shown” to “shown here”.

Page 13 Line 296 – change “resulted in” to “required”.

Page 14 Line 304 – change “plan properly” to “properly plan”.

Page 14 Line 305 – change “performance goal” to “performance objectives”.

Page 14 Line 305 – change “larger” to “higher”.

Page 14 Lines 306-308 – I think the Authors need to acknowledge that this is an entirely obvious and expected statement. Perhaps they can insert the word “unsurprisingly” or something similar.

Page 14 Line 214 – change this sentence “Therefore, manipulating properly the ApP allows managing the training loads to be performed using SSGs…” to “Therefore, manipulating ApP allows for training loads in SSGs to be managed…”

Page 15 Line 336 – change “smaller” to “lower”.

Page 15 Line 340 – as well as suggesting that “these positions should be conditioned differently…”, I wonder whether the Authors should also discuss that perhaps the position-specific conditioning stimulus varies according to the player density. Furthermore, just because the movement characteristics don’t replicate the match, that does not mean that there is no conditioning stimulus – it just means that it is not exactly the same as the match.

Page 15 Line 340 – change “used” to “tend”.

Page 15 Line 341 – change “…while central-midfielders and forwards move across the pitch to gain possession of the ball…” to “…while central-midfielders and forwards tend to cover a greater area of the pitch in order to gain possession of the ball…”

Page 16 Line 352 – I think the Authors need to elaborate on point iv) more, rather than just mention it at the very end of the Discussion.

Conclusion

Page 16 Line 363 – I am not sure what the Authors mean by “more physiologically based exercises”. The entire premise of their study is to assess the physiological characteristics of SSGs, so it is contradictory to imply that these are not activities are not physiologically based. The Authors need to reconsider their terminology – and thinking – here.

Figure

Figure 1 – It is not clear exactly what information or concept the Authors are trying to present in Figure 1, particularly given the multiple variables all listed on the y-axis. Furthermore, there is no mention of Figure 1 in the text.

Figures 2 & 3 – I can see no reason why the Metabolic Power results are presented in a separate figure, rather than being an additional panel within Figure 2. For clarity and also ease of comparison, it would make sense that Figure 2 comprises 5 panels (A-E) representing each of the 5 variables assessed.

6. PLOS authors have the option to publish the peer review history of their article (what does this mean?). If published, this will include your full peer review and any attached files.

Reviewer #1: No

Reviewer #2: No

Reviewer #3: Yes: Ted Polglaze

---

## [Author Response · Author response to Decision Letter 0]

21 Apr 2020

Journal Requirements:

As by the request from the Editorial Office, we revised style requirements. We hope this new version could reach the desired quality. Please see text.

2. Thank you for including your ethics statement; "All participants gave their written consent after a full explanation of the purpose of the study and the experimental design. The local University Ethics Committee approved the study and was performed in accordance with the principles of the Declaration of Helsinki (1975). "

As by the request from the Editorial Office, we have amended it in this new version of the manuscript. Please see text. 

Additional Editor Comments (if provided):

Please, address point-to-point all reviewers' issues, in particular Reviewer 3's ones. In its current format MS is not suitable for publication.

We thank the Editor for the opportunity to revise our manuscript. We hope this new version could reach the desired quality. 

Reviewer #1: 

I congratulate the authors for the excellent and innovative work presented. This represents a huge effort and dedication compensated with outstanding results for better knowledge and application of SSG. 

We thank the Reviewer. 

Reviewer #2: 

The purpose of this study was to determine the area-per-player during small- or large-sided games with or without goalkeeper that replicates the relative (m�min-1) total distance, high-intensity running distance, sprint distance and metabolic power covered during official matches. The paper is very interesting, nevertheless, I would like to suggest some ideas in order to clarify the reading of the text. I hope with my comments the quality of the study can be improved.

We thank the reviewer for her/his suggestions, and we hope this new version could reach the desired quality. 

Specific comments:

Introduction

The main idea that is discussed in the introduction is related to compare physical demands between small-sided games played with and without goalkeepers. In this line, I believe you need to include all references that study this comparison. Here some of them:

• Sassi, Reilly, & Impellizzeri (2004)

• Mallo, J., y Navarro, E. (2008). Physical load imposed on soccer players during small-sided training games. Journal of Sports and Physical Fitness, 48(2), 166-171. 

• Casamichana, D., Castellano, J., González-Morán, A., García-Cueto, H., y García-López, J. (2011). Demanda fisiológica en juegos reducidos de fútbol con diferente orientación del espacio. Revista Internacional de Ciencias del Deporte, 23(7), 141-154.

• Köklü Y, Sert O, Alemdaroğlu U, & Arslan Y. (2015). Comparison of the physiological responses and time motion characteristics of young soccer players in small sided games: the effect of goalkeeper. Journal of Strength and Conditioning Research, 

• Joaquín González-Rodenas1, Ferran Calabuig1, Rafael Aranda1. Effect of the Game Design, the Goal Type and the Number of Players on Intensity of Play in Small-Sided Soccer Games in Youth Elite Players. Journal of Human Kinetics volume 49/2015, 229-235

According to the reviewer’s suggestion, we have deepened this part, citing some of the suggested articles. Please see text and references. 

Methods

Procedure

Please clarify (maybe including interchangeability formulas) more information about interchangeability.

Following both reviewers’ suggestions, we have populated this part. However, we think this is more suitable within the results section, where an entire sub-section has been dedicated. Please see text. 

Line 166, than or then?.

This was “then”. We have reworded it, please see text. 

Line 169 please specify what the system used to register each variable.

Both GPS and MCIS were used simultaneously. This has been stated, please see text. 

Lines 172-173, about calibration equations, were they calculated or copied from Buccheit’s study. Please clarify.

The calibration equations were calculated and are shown within the new figure 2. 

Statistical analysis

Could you include Bland and Altman plot statistics to determine the relationship between the GPS and the MCIS?

As requested, the new figure 2 shows both the Bland-Altman analysis and correlation between GPS vs MCIS for each dependent parameter. 

Results

I have tried to put the values appear in the table 1 in the figures A, B, C and D, but they do not match in the regression lines, am I doing something wrong? (e.g., the total value for all position of 115 m in SSG-P or 187 in SSG-G added to the figure A do not match with the reference line of the game and regression lines). Please explain.

We understand this specific reviewer’s concern. As specified, the total values in table 1 is the mean of the individual values of the individual intersection points. In the figures, each black or white dot is the mean of the relative distance recorded during each SSGs per each player, averaging one figure for each player. Please keep in mind that the current investigation derives from real-life elite soccer, where all players do not perform the same activities. In this real-life context, some players could have missed some specific SSG session, resulting in a possible mismatching between the values indicated in table-1 and the intercept drawn in figure 3. This has been clearly shown in Supplementary table 1 and 2. However, we acknowledge that this could generate some confusion, so we have removed the intercepts from figure 3. 

Discussion

Please remove all the sections inside this section (e.g., Methodological consideration”, “main findings”…). 

As required, we have removed the sub-headings. 

All information about methodological consideration must include like limitations of the study, in the last part of the discussion section.

As suggested, we have included a dedicated paragraph concerning the limitations of the study. Please see text. 

Other limitation that the authors must considerer include could be that no reference about number of players per team used in the tasks. It is not the same to play a 4:4 in a relative space of 100 m per player, that one 8:8 in the same relative space per player.

Actually, the specific reference for the number of players and the total area is reported in supplementary table 1 and 2. However, the possible difference exemplified by the reviewer has been taken into account showing the confidence interval bands of each regression line in figure 3. We would like to remind that figure 3 shows the mean of each SSG regression line.

Line 330, freer?¿?¿?

That was a typo. We have reworded into “free”. Please see text.

Please add to the discussion section the connection with the papers suggested in the introduction.

As suggested, we have added a connection with those papers. Please see text. 

The idea argued in this paragraph (from line 318 to 332) could be improved adding information that from my point of view is important. Because previous academic literature (Castellano et al., 2013) suggest the variables regarding to acceleration are more demanded in oriented games. So depending of the variable considered the task could be more or less demanded. The idea behind this more stimulation of accelerations in task with goalkeepers is that players need to overpass opponent or defensive lines in order to achieve the rival goal. Maybe could be interesting to include this information and discuss about it.

As suggested, we have populated this part. Please see text. 

The last section, conclusion, I think could be interesting how the coaches could apply the result suggested in this study. Different position needs no same ApP, but how can coaches to design game tasks following the results. I think include this applicability is key.

We agree with the reviewer and we have revised this part accordingly. Please see text. 

Reviewer #3: 

General Comments

I think the basic premise of this study is sound, and the data potentially provides meaningful and interesting information for sport science researchers and practitioners. However, I feel there are a few important issues that the Authors need to consider in revising this manuscript.

We thank the reviewer for his suggestions, and we hope that this new version could reach the desired quality. 

Firstly, the whole premise of that study is to compare various SSG formats to match conditions, and yet the measurement of match conditions is with an entirely different measurement system. It is not sufficient to apply an existing “calibration equation”, as this was developed from entirely different systems. If the Authors want to compare data between their current systems, they need to, at the very least, present some pilot data that compares the output from their own GPS & MCIS products.

We apologize for not having provided a thorough comparison between these GPS vs MCIS products in the previous version. In this new version, we have included the Bland-Altman analysis, the correlation graphs and the calibration equation for each dependent parameter. Actually, the calibration equations used in the previous version were “our” equations, but we acknowledge that this was not specified. Please see the new results section. 

Secondly, the Authors seem to rely on 2 articles which question the validity of the metabolic power model whilst ignoring subsequent research which supports its application in team sports (and addresses some of the misconceptions and flaws of those early papers. Meanwhile, their own results seem to indicate that metabolic power is an appropriate tool for monitoring the demands of SSGs, but the Authors appear to overlook this finding.

We agree with the reviewer. Following this specific suggestion, we have accordingly rewritten part of the introduction and discussion. Please see text. 

Thirdly, the Authors seem to focus too much on whether a particular SSG replicates the match demands, and neglect to adequately address some of the other objectives and advantages of SSGs. At a very simplistic level, if the intent is merely to replicate the demands/characteristics of a match, then as a reader I find myself asking “why not just play matches then?”. The Authors need to provide more context and perspective regarding the role of SSGs in training.

We agree with the reviewer. Following this suggestion, we have implemented and contextualized this topic within the discussion section. Please see text. 

Specific Comments

Introduction

Page 3 Line 52 – I suggest this sentence be re-worded to read “To assess these demands, contemporary player-tracking technologies such as global positioning system (GPS) or semi-automatic video-based multi-camera image systems (MCIS), are typically used.” Note that “system” should be singular.

As suggested, we have rephrased the sentence. Please see text. 

Page 3 Line 61 – there are numerous issues with the term “MechW” mentioned here. Firstly, when the authors state that acceleration and deceleration is “often referred to as mechanical work”, the paper that they cite (Gaudino 2014) never uses the term nor the acronym. Secondly, and more importantly, by defining that MechW applies to changes in speed, this implies that no mechanical work is done when running at a constant speed. This is clearly not the case. The Authors need to abandon the term MechW throughout the manuscript and adopt more appropriate terminology. Just as importantly, this needs to filter through to the way the Authors interpret and discuss their findings. 

We agree with the reviewer. For this reason, we have replaced “MechW” with “Acc/Dec” throughout the whole manuscript. Please see text. 

Page 3 Line 66 – change “reviewed” to “found”.

As suggested, we have reworded it. Please see text. 

Page 3 Line 69 – change “Over the last years” to “In recent times”.

As suggested, we have reworded it. Please see text.

Page 4 Line 75 – the statement “it does not take into account the individual running economy” is a throw-away line that lacks evidential support. Neither of the papers cited here actually assessed individual economy, and in any case it is the COST rather than ECONOMY that is used to estimate metabolic power. Meanwhile, there is substantial experimental evidence in the literature that the cost of locomotion varies little between individuals.

We agree with the reviewer and we have removed this specific statement. Please see text. 

Page 4 Line 77 – the statement here indicates a misunderstanding of the metabolic power model, as it implies that it is only relevant for assessing accelerations and decelerations but not high-speed running. This represents one of many misconceptions from previous literature that the Authors are perpetuating.

As also previously suggested, we have rewritten this whole part. Please see text. 

Material and Methods

Page 5 Line 98 – remove “the” prior to “data collection”.

As suggested, we have reworded it. Please see text.

Page 5 Line 99 – remove “the” prior to “data collection”.

As suggested, we have reworded it. Please see text.

Page 5 Line 100 – change “…full in-group re-integration.” to “…their return to full training.”

As suggested, we have reworded it. Please see text.

Page 5 Line 106 – change “…during the in-season period (August 2014 – May 2016)” to “during the competition period across two seasons (August 2014 – May 2016) ”.

As suggested, we have reworded it. Please see text.

Page 5 Line 108 – change “All sessions were performed on two grass pitches preserved by qualified operators and performed at the same time to limit the effects of circadian variations” to “All sessions were performed on two grass pitches preserved by qualified operators and were conducted at the same time of day to limit the effects of circadian variation ”. Note here that “variation” is singular.

As suggested, we have reworded it. Please see text.

Page 5 Line 118 – change “S1 and S2 Tables” to “Tables S1 and S2”.

As suggested, we have reworded it. Please see text.

Page 5 Line 119 – change “Both small- or large-sided games was abbreviated…” to “Both small- and large-sided games were abbreviated…”

As suggested, we have reworded it. Please see text.

Page 6 Line 124 – change “…warm-up under the club’s staff guidance” to “warm-up under the guidance of club staff”.

As suggested, we have reworded it. Please see text.

Page 7 Line 153 – change “each data collection” to “each session”.

As suggested, we have reworded it. Please see text.

Page 7 Line 153 – change “allow the acquisition” to “allow for acquisition”.

As suggested, we have reworded it. Please see text.

Page 7 Line 154 – change “every player wore the same device for each training session” to “each player wore the same unit for every training session”. Note that “unit” should be used instead of “device”.

As suggested, we have reworded it. Please see text.

Page 8 Line 173 – the “calibration equations” proposed by Buchheit et al. are specific to the devices and tracking systems assessed in that study. There is no reason to assume that they would equally apply to the GPS and MCIS brands used in the present study, and therefore makes the use of this equation dubious.

As also described above, we failed to report our “calibration equations”, on whose basis the present results were derived from. In this new version, we have provided the current “calibration equations” for each dependent parameter. Please see the results section.

Discussion

General Comments – in addition to determining what ApP values replicate match demands for each position, I think it is important that the Authors acknowledge (and discuss) that SSGs also have a role in overloading different aspects of physical demands as well as emphasising various tactical or technical parameters. If the only objective of SSGs was to replicate match demands, then surely teams would just play matches. I think the Authors need to include this point in their interpretation and discussion of the results.

We definitely agree with the reviewer. Following this suggestion, we have implemented and contextualized this topic within the discussion section. Please see text. 

Page 13 Line 273 – this first sentence is a bit clumsy with respect to wording. I suggest something like “In order to properly interpret the results observed in the present study, some preliminary considerations need to be taken into account”.

That sentence has been removed and that section has been moved to the end of the discussion as “limitations section”, in accordance with the reviewer’s-2 suggestions. 

Page 13 Line 281 – change “amount” to “magnitude”.

As suggested, we have reworded it. Please see text.

Page 13 Line 284 – again here the Authors are perpetuating a misconception surrounding the metabolic power model, and ignoring more recent and relevant literature which demonstrates that the model is quite accurate in estimating the energetic cost of LOCOMOTION for variable-speed activity (e.g. Osgnach & di Prampero, 2018)

As suggested, we have rewritten this part. Please note that we have moved it at the end of the discussion. Please see text. 

Page 13 Lines 286-288 – I think this is an example of where the Authors rely too much on one or two papers rather than providing a more extensive and balanced assessment of existing literature.

As suggested, we have rewritten this part. Please note that we have moved it at the end of the discussion. Please see text.

Page 13 Line 293 – change “here shown” to “shown here”.

As suggested, we have reworded it. Please see text.

Page 13 Line 296 – change “resulted in” to “required”.

As suggested, we have reworded it. Please see text.

Page 14 Line 304 – change “plan properly” to “properly plan”.

As suggested, we have reworded it. Please see text.

Page 14 Line 305 – change “performance goal” to “performance objectives”.

As suggested, we have reworded it. Please see text.

Page 14 Line 305 – change “larger” to “higher”.

As suggested, we have reworded it. Please see text.

Page 14 Lines 306-308 – I think the Authors need to acknowledge that this is an entirely obvious and expected statement. Perhaps they can insert the word “unsurprisingly” or something similar.

As suggested, we have reworded it. Please see text.

Page 14 Line 314 – change this sentence “Therefore, manipulating properly the ApP allows managing the training loads to be performed using SSGs…” to “Therefore, manipulating ApP allows for training loads in SSGs to be managed…”

As suggested, we have reworded it. Please see text.

Page 15 Line 336 – change “smaller” to “lower”.

As suggested, we have reworded it. Please see text.

Page 15 Line 340 – as well as suggesting that “these positions should be conditioned differently…”, I wonder whether the Authors should also discuss that perhaps the position-specific conditioning stimulus varies according to the player density. Furthermore, just because the movement characteristics don’t replicate the match, that does not mean that there is no conditioning stimulus – it just means that it is not exactly the same as the match.

As suggested, we have rephrased the sentence and explained more in depth the between-position differences and the possible consequences. Please see text. 

Page 15 Line 340 – change “used” to “tend”.

As suggested, we have reworded it. Please see text.

Page 15 Line 341 – change “…while central-midfielders and forwards move across the pitch to gain possession of the ball…” to “…while central-midfielders and forwards tend to cover a greater area of the pitch in order to gain possession of the ball…”

As suggested, we have reworded it. Please see text.

Page 16 Line 352 – I think the Authors need to elaborate on point iv) more, rather than just mention it at the very end of the Discussion.

As suggested, we have elaborated this within the “position” paragraph. Please see text. 

Conclusion

Page 16 Line 363 – I am not sure what the Authors mean by “more physiologically based exercises”. The entire premise of their study is to assess the physiological characteristics of SSGs, so it is contradictory to imply that these are not activities are not physiologically based. The Authors need to reconsider their terminology – and thinking – here.

We definitely agree that this terminology was not correct, and we have removed that statement. Please see text. 

Figure

Figure 1 – It is not clear exactly what information or concept the Authors are trying to present in Figure 1, particularly given the multiple variables all listed on the y-axis. Furthermore, there is no mention of Figure 1 in the text.

Actually, figure 1 was conceived to have a graphical explanation of the study design. We prefer maintaining it; however, we have amended it following this specific suggestion. We have now mentioned in the text, within the methods section. 

Figures 2 & 3 – I can see no reason why the Metabolic Power results are presented in a separate figure, rather than being an additional panel within Figure 2. For clarity and also ease of comparison, it would make sense that Figure 2 comprises 5 panels (A-E) representing each of the 5 variables assessed.

As suggested, we have merged the figures 2 and 3 into a new single figure 3. Please see the results section.

---

## [Decision Letter · Decision Letter 1]

28 Aug 2020

PONE-D-20-02888R1

Area per player in small-sided games to replicate the external load and estimated physiological match demands in elite soccer players

PLOS ONE

Dear Dr. RIBOLI,

Thank you for submitting your manuscript to PLOS ONE. After careful consideration, we feel that it has merit but does not fully meet PLOS ONE’s publication criteria as it currently stands. Therefore, we invite you to submit a revised version of the manuscript that addresses the points raised during the review process.

Please, one further effort to address remaining Reviewer 3's issues.

We look forward to receiving your revised manuscript.

Kind regards,

Luca Paolo Ardigò, Ph.D.

Academic Editor

PLOS ONE

Additional Editor Comments (if provided):

Please, one further effort to address remaining Reviewer 3's issues.

Reviewers' comments:

Reviewer's Responses to Questions

**Comments to the Author**

1. If the authors have adequately addressed your comments raised in a previous round of review and you feel that this manuscript is now acceptable for publication, you may indicate that here to bypass the “Comments to the Author” section, enter your conflict of interest statement in the “Confidential to Editor” section, and submit your "Accept" recommendation.

Reviewer #3: All comments have been addressed

Reviewer #4: All comments have been addressed

2. Is the manuscript technically sound, and do the data support the conclusions?

Reviewer #3: Yes

Reviewer #4: Yes

3. Has the statistical analysis been performed appropriately and rigorously? 

Reviewer #3: Yes

Reviewer #4: Yes

4. Have the authors made all data underlying the findings in their manuscript fully available?

Reviewer #3: Yes

Reviewer #4: Yes

5. Is the manuscript presented in an intelligible fashion and written in standard English?

Reviewer #3: Yes

Reviewer #4: Yes

6. Review Comments to the Author

Reviewer #3: General Comments

I commend the Authors for their thorough and well-considered revision. I have only minor corrections, but believe that this manuscript is suitable for publication.

Specific Comments

Introduction

Page 3 Line 70 – change “controversial” to “conflicting”.

Page 4 Line 75 – overall, I commend the Authors for providing a more balanced (and evidence-based) account of Metabolic Power. However, in saying that the model is “…used to estimate the energy cost of high-intensity running activities…”, this implies that it is only useful for this purpose, whereas in fact it applies across all intensities. So, please change “the energy cost of high-intensity running activities in team sports” to “the energy cost of locomotion in team sports”.

Reviewer #4: All my comments have been satisfactorily adressed. In my opinio, the manuscript can be opublished in its current form.

7. PLOS authors have the option to publish the peer review history of their article (what does this mean?). If published, this will include your full peer review and any attached files.

Reviewer #3: **Yes: **Ted Polglaze

Reviewer #4: No

---

## [Author Response · Author response to Decision Letter 1]

29 Aug 2020

PONE-D-20-02888R1

Area per player in small-sided games to replicate the external load and estimated physiological match demands in elite soccer players

PLOS ONE

Additional Editor Comments (if provided):

Please, one further effort to address remaining Reviewer 3's issues.

Reviewer #3: General Comments

I commend the Authors for their thorough and well-considered revision. I have only minor corrections, but believe that this manuscript is suitable for publication.

We would to thank the Reviewer for his useful comments that help us to improve the manuscript.

Specific Comments

Introduction

Page 3 Line 70 – change “controversial” to “conflicting”.

As suggested, we changed “controversial” to “conflicting”. Please see text.

Page 4 Line 75 – overall, I commend the Authors for providing a more balanced (and evidence-based) account of Metabolic Power. However, in saying that the model is “…used to estimate the energy cost of high-intensity running activities…”, this implies that it is only useful for this purpose, whereas in fact it applies across all intensities. So, please change “the energy cost of high-intensity running activities in team sports” to “the energy cost of locomotion in team sports”.

We thank the Reviewer for his comment. We strongly agree with him and we reworded the sentence. Please see text.

Reviewer #4: All my comments have been satisfactorily adressed. In my opinio, the manuscript can be opublished in its current form.

We would to thank the reviewer for his/her comment and for his/her previous suggestions that help us to improve the manuscript.

---

## [Decision Letter · Decision Letter 2]

4 Sep 2020

Area per player in small-sided games to replicate the external load and estimated physiological match demands in elite soccer players

PONE-D-20-02888R2

Dear Dr. RIBOLI,

We’re pleased to inform you that your manuscript has been judged scientifically suitable for publication and will be formally accepted for publication once it meets all outstanding technical requirements.

Kind regards,

Luca Paolo Ardigò, Ph.D.

Academic Editor

PLOS ONE

Additional Editor Comments (optional):

Congratulations for the good job.

Reviewers' comments:

Reviewer's Responses to Questions

**Comments to the Author**

1. If the authors have adequately addressed your comments raised in a previous round of review and you feel that this manuscript is now acceptable for publication, you may indicate that here to bypass the “Comments to the Author” section, enter your conflict of interest statement in the “Confidential to Editor” section, and submit your "Accept" recommendation.

Reviewer #3: All comments have been addressed

2. Is the manuscript technically sound, and do the data support the conclusions?

Reviewer #3: Yes

3. Has the statistical analysis been performed appropriately and rigorously? 

Reviewer #3: Yes

4. Have the authors made all data underlying the findings in their manuscript fully available?

Reviewer #3: Yes

5. Is the manuscript presented in an intelligible fashion and written in standard English?

Reviewer #3: Yes

6. Review Comments to the Author

Reviewer #3: (No Response)

7. PLOS authors have the option to publish the peer review history of their article (what does this mean?). If published, this will include your full peer review and any attached files.

Reviewer #3: **Yes: **Ted Polglaze

---

## [Editor Report · Acceptance letter]

14 Sep 2020

PONE-D-20-02888R2 

Area per player in small-sided games to replicate the external load and estimated physiological match demands in elite soccer players 

Dear Dr. Riboli:

I'm pleased to inform you that your manuscript has been deemed suitable for publication in PLOS ONE. Congratulations! Your manuscript is now with our production department. 

Kind regards, 

on behalf of

Dr. Luca Paolo Ardigò 

Academic Editor

PLOS ONE